# Value-Added Products from Coffee Waste: A Review

**DOI:** 10.3390/molecules28083562

**Published:** 2023-04-18

**Authors:** Yoon-Gyo Lee, Eun-Jin Cho, Shila Maskey, Dinh-Truong Nguyen, Hyeun-Jong Bae

**Affiliations:** 1Bio-Energy Research Center, Chonnam National University, Gwangju 500-757, Republic of Korea; 2School of Biotechnology, Tan Tao University, Duc Hoa 82000, Long An, Vietnam; 3Department of Bioenergy Science and Technology, Chonnam National University, Gwangju 500-757, Republic of Korea

**Keywords:** coffee waste, value-added products, spent coffee grounds, bioactive compounds, bio-sugars, bio-oils, organic acids, biopolymer

## Abstract

Coffee waste is often viewed as a problem, but it can be converted into value-added products if managed with clean technologies and long-term waste management strategies. Several compounds, including lipids, lignin, cellulose and hemicelluloses, tannins, antioxidants, caffeine, polyphenols, carotenoids, flavonoids, and biofuel can be extracted or produced through recycling, recovery, or energy valorization. In this review, we will discuss the potential uses of by-products generated from the waste derived from coffee production, including coffee leaves and flowers from cultivation; coffee pulps, husks, and silverskin from coffee processing; and spent coffee grounds (SCGs) from post-consumption. The full utilization of these coffee by-products can be achieved by establishing suitable infrastructure and building networks between scientists, business organizations, and policymakers, thus reducing the economic and environmental burdens of coffee processing in a sustainable manner.

## 1. Introduction

The recent increases in solid waste production, including agricultural, industrial, household, human, and animal waste, have become an important problem that causes environmental pollution worldwide. In the face of these challenges, waste valorization employing biotechnological approaches is becoming a sustainable green approach for solving this problem [1]. Coffee is one of the most consumed beverages in the world and is the second-largest traded commodity after petroleum, which highlights its immense global market share [2]. However, the coffee industry generates large amounts of toxic solid waste residues that cause serious environmental issues, and, therefore, additional efforts are needed to develop sustainable solutions [3,4,5,6,7].

Each year, the coffee industry generates over 10 million tons of coffee waste worldwide [8]. The waste consists of husks, pulp, mucilage, silverskins, and spent coffee grounds (SCGs), which are derived from the different steps involved in coffee processing, including harvesting, processing, roasting, and brewing [9]. Among these waste materials, SCGs (0.6 tons per ton of coffee) represent the most abundant residue generated during processing, whereas coffee pulp and husks account for only 0.50 and 0.18 tons per ton of fresh coffee, respectively [10,11]. In Korea, the generation of SCGs has been increasing every year, starting from 93,397 tons in 2012 to 149,038 tons on a dry weight (DW) basis in 2019 [12].

The chemical components of coffee waste vary considerably depending on the coffee variety processing steps, and type of by-products [13]. Many studies have reported on the variation of the chemical components of coffee waste according to the coffee species. There are more than 120 species of the genus Coffea belonging to the family Rubiaceae. However, only the arabica and robusta varieties, which, respectively, derive from the *C. arabica* and *C. canephora* species, have been commercially produced and distributed worldwide [14,15,16]. A study comparing the composition of aqueous extracts from arabica and robusta coffee pulp revealed that the arabica variety processes higher antioxidant levels [17]. However, upon comparing the antioxidant activities of coffee silverskin from both species, the silverskin from robusta species was found to have higher antioxidant activity [18]. The superior antioxidant efficacies of robusta species were also reported by a study that characterized green coffee extract. Antioxidant activity could be affected not only by the coffee species, but also the processing, extraction, and brewing method [19,20]. Additionally, previous studies have also demonstrated that different coffee processing methods can alter the characteristics coffee waste. For example, dry and wet processes can be applied to coffee cherries to remove pulps, husks, and silverskins for the production of green coffee beans as shown in Figure 1 [21]. Typically, *C. robusta* and *C. arabica* are processed via the dry and wet methods, respectively. In the dry method, the coffee cherries are dried before removing other components, whereas the cherries are kept in water in the wet method. Importantly, the latter method promotes microbial fermentation, which produces superior quality and aroma. Although the robusta variety typically has a higher caffeine content compared to arabica, the quality of arabica coffee is generally considered to be superior compared to that of robusta. Therefore, arabica coffee accounts for the overwhelming majority of the global coffee market share (75%) [14].

The growth and expansion of the coffee processing industry have led to an annual increase in coffee waste, which is typically discarded in landfills, mixed with animal fodder, or incinerated [22]. In response to these challenges, transforming these waste products into animal feed and fertilizer, or using biotechnology to convert them into biofuels, enzymes, and aroma compounds are promising approaches to sustainably solve this issue. However, several studies have demonstrated that coffee waste is unsuitable for the production of animal feed due to its caffeine, tannin, and alkaloids contents, which not only affects animal health, but also diminishes the palatability of the diets [23,24]. Furthermore, due to its high content of tannins and caffeine, coffee waste can degrade soil quality, induce cytotoxicity, hamper microbial activity during biotechnological transformation, trigger physiological changes in the central nervous systems of cattle and fish, and induce carcinogenicity in animals when the waste is used as a component of animal feed [25,26,27]. Another study reported that the presence of chlorogenic acids in coffee waste can interfere with seed germination and growth [11]. Nevertheless, the use of coffee by-products as functional ingredients has been an emerging field in the food industry due to the high concentration of antioxidant compounds in coffee. Moreover, approximately 0.5 and 0.2 metric tons of pulp and husk per metric ton of fresh coffee are produced during coffee processing, respectively. Coffee waste has been utilized to produce bio-sugar, biofuel, fertilizer, enzymes, dietary fiber, and bioactive compounds [28]. Therefore, the by-products of coffee processing are no longer considered waste but are instead seen as a promising material for the production of many value-added products [29,30].

This review provides an up-to-date summary of value-added products obtained not only from coffee processing by-products (coffee pulps, husks, silver skins, and SCGs) but also from coffee flowers and leaves. Finally, this review describes various by-products and their applications, as well as the promising prospects of each coffee-derived product.

## 2. Components of Coffee By-Products and Their Chemical Compositions

### 2.1. Coffee Leaves

Coffee beans have been consumed as a beverage since the mid-15th century. However, until relatively recently, coffee leaves had not been widely used. The application of coffee leaf extracts as a beverage has a long history in certain localities where coffee has been planted for centuries. The human health benefits of the coffee leaves have recently garnered increasing attention due to their abundant content of bioactive components. However, very few studies have characterized the bioactive properties of the phytochemical compounds derived from coffee leaves. The most abundant bioactive compound in coffee leaves is caffeine, with a concentration of approximately 24.5 g/kg of dried leaves [31,32]. In fact, due to their high caffeine content, old coffee leaves may harm the soil and the beneficial microorganisms living in the soil and coffee roots, particularly in large-scale coffee farming areas. Therefore, collecting coffee leaves for further applications not only promotes the utilization of this resource but also reduces the environmental impacts of coffee leaf waste. Despite being commonly considered a by-product, coffee leaves are a rich source of antioxidants, particularly mangiferin and hydroxycinnamic acid esters (HCEs), and have, therefore, been extensively studied due to their numerous potential health benefits [33]. Nevertheless, consuming coffee leaves in beverages is not advisable due to the negative effects of high caffeine ingestion. Moreover, a total of 47 different volatile compounds have been identified in essential oils from coffee leaves with (E)-2-hexenal (39.7%) and 1-hexanol (32.1%) being among the most abundant. Importantly, these coffee leaf-derived oils can be used as an alternative to herbal essential oils [34]. Moreover, a recent study tested the applicability of coffee leaves for various purposes, including ethnomedicine, facial cleansers, tobacco substitutes, animal feed, proliferating agents, packaging materials, absorbance pads, and deodorizers [35].

### 2.2. Coffee Flowers

Similar to coffee leaves, few studies have explored the characteristics and application of coffee flowers. However, several studies have reported on the physiological effects of coffee fruits and beans. An adult coffee tree is estimated to produce approximately 30,000–40,000 flowers annually. Therefore, value-added products derived from coffee flowers could be profitably produced, provided that the flowers can be harvested without affecting the production yield of coffee cherries [15]. Coffee flowers are a known source of caffeine and trigonelline, as well as volatile components such as epoxygeraniols (2,3-epoxygeraniol and 6,7-epoxygeraniol) and epoxynerol [36,37]. A previous study reported that n-pentadecane is the most abundant compound in coffee flowers, followed by 8-heptadecane and geraniol [38], making coffee flowers an alternative source for the production of distinctive fragrances. Like other flowers, coffee flowers also contain numerous components such as carbohydrates, caffeine, tannins, and polyphenols. In our previous studies, a green approach based on pressurized hot water extraction was used to obtain bioactive compounds, such as melanoidins and biosugars from coffee flowers. Interestingly, our finding also confirmed that caffeine and trigonelline were the main compounds extracted from coffee flowers, with yields of 1070.8 mg and 1092.8 mg/100 g DW, respectively. These findings suggested that coffee flowers are a promising option in the beverage industry, which could increase the economic benefits of coffee farmers [39]. The flowers can also be used to brew a tea-like infusion with hot water. Another secondary product in coffee plantations is a type of mono floral honey produced from coffee flowers, which is rarely available due to the short flowering period of coffee flowers [40].

### 2.3. Coffee Pulps and Husks

The by-products generated during each step of coffee processing are presented in Figure 1. In both the dry and wet processing methods, coffee pulp (CP) and husk (CH) account for high proportions of the overall coffee cherry biomass on a dry weight (DW) basis, reaching an estimated 0.5 and 0.2 metric tons per metric ton of fresh coffee, respectively [41]. In addition to carbohydrates, minerals, and proteins, CP and CH contain high levels of caffeine, chlorogenic acid, and tannins, which limits their use as animal feed or as fertilizer due negative effects of these compounds on animal health and seed germination and plant growth. However, CP and CH biomass has long been used for biogas production [42]. Additionally, these by-products are also used for mushroom cultivation, wherein 73% of the substrate is utilized and the remaining amount can be used as fertilizer [43,44]. CP and CH are also useful for the production of bioethanol, biofuels, enzymes, and bioactive compounds [11].

For enzyme production, a study achieved maximum xylanase activity rates of 9.475 U/g via solid-state fermentation (SSF) of coffee husks as the sole carbon source using *Penicillium* sp. [2]. A previous study reported that SSF of CP after alkali pretreatment using *Acinetobacter* sp. produced 888 U/mL of cellulase after 60 h of incubation [45]. β-Glucosidase is another important enzyme required for the degradation of lignocellulose. A previous study reported that this enzyme was produced by *Bacillus substilis* CCMA 0087 using CPs as a substrate, with a maximum enzyme yield of 22.59 UI/mL after 24 h [46]. Additionally, bioethanol is a major byproduct obtained from CP and CH. Before saccharification and fermentation, CP and CH are commonly pretreated with alkali or dilute acid. Some studies have reported encouraging results, such as ethanol yields of approximately 70 g/L after 24 h of fermentation of a hydrolysate mixture of sugarcane molasses and CPs [47], as well as 13.66 g/L of ethanol from alkaline-pretreated CP alone [48]. A particular pretreatment method known as popping pretreatment can enhance the saccharification of individual agricultural biomass or some mixtures (including CH, coconut coir, and cassava stems). Biomass pretreated using this method has been reported to yield up to 55.2 g DW bioethanol per 300 g DW of mixed biomass, which corresponds to a conversion rate of 77.3% [49]. From an economic perspective, a techno-economic assessment demonstrated that the net value of the production of bioethanol derived from CPs justified its use as an economically feasible energy source, in addition to offsetting CO_2_ emissions to adapt to environmental requirements [50].

CPs and CHs contain high levels of chlorogenic and caffeic acids. Using an inexpensive enzymatic extraction method, chlorogenic acid was the main product (36.1%) of SSF with *Aspergillus tamarii*, *Rhizomucor pusillus*, and *Trametes sp*., followed closely by caffeic (33%) acids [51]. Moreover, in an earlier study, the enzymatic extraction of 1 kg of CPs rendered a total of 5.4 g of ferulic, caffeic, p-coumaric, and chlorogenic acids [52]. In vitro experiments have demonstrated the antioxidant, antibacterial, and anti-inflammatory properties of CPs, which were attributed to the chlorogenic and caffeic acids contents [53,54]. Lactic acid yields of 55.5–67.6 kg DW per 1000 kg DW of CPs have also been achieved via fermentation with *Bacillus coagulans* at the pilot scale [55]. Several CH-derived products such as activated carbon powder and dietary fiber supplements have also attracted substantial attention [56,57,58]. Interestingly, due to their rich phenolic contents, CFs can be used to produce cascara beverages [59], which have already been widely commercialized in the coffee beverage market. Additionally, CHs are used in the production of bio-pesticides by fermentation with *Bacillus sphaericus* and *B. thuringiensis* subsp. *israelensis* to control disease-transmitting mosquitoes [60].

### 2.4. Coffee Silverskin

Coffee silverskin, which is produced during the roasting process, only accounts for approximately 4.3% (*w*/*w*) of the coffee cherry, but it is rich in dietary fiber (80%), antioxidants, and phenolic compounds [2,61,62]. The skin can also be used as raw material to extract antioxidants due to its high chlorogenic acid content [63,64]. A study reported that polysaccharides in the form of cellulose and hemicellulose are the most abundant components in coffee silverskin, accounting for 40.5% (*w*/*w*) of its total mass on a DW basis [61]. Previous studies have also reported the production of bio-butanol from coffee silverskins. However, the use of coffee silverskin and SCGs for bio-butanol production can be prohibitively difficult, especially when compared to other raw lignocellulosic materials, which are widely available in large amounts [65,66]. Additionally, bio-H_2_ production of coffee silverskin using *E. coli* with the yield of 2.15 mL/g raw coffee silverskin was reported by Trchouian group [67,68].

### 2.5. SCGs

SCGs are obtained after brewing a coffee or during the production of instant coffee, accounting for approximately 50% DW of the coffee cherry biomass. The global annual production of SCGs was estimated at 6 million metric tons, which corresponds to 650 kg of SCGs per metric ton of green coffee beans [10]. The reuse of SCGs has gained particular attention in recent years because these residues must be treated prior to being discharged into the environment due to their potential ecological impacts. Additionally, SCGs contain high amounts of organic compounds including carbohydrates, lipids, and proteins, as well as phenolic and bioactive compounds such as caffeine, chlorogenic, caffeic acids, cafestol, and kahweol, and can be easily collected in large amounts from coffee factories and shops [5,69]. Therefore, substantial efforts have been made to convert SCGs into value-added products such as biofuels, bioactive compounds, and biomaterials.

Compared to other lignocellulosic materials (lignin, cellulose, and hemicellulose), SCGs contain a high content of hemicelluloses (30–40 wt%) and lignin (25–30 wt%) and relatively small amounts of cellulose (approximately 8.6–13.3 wt%) [70]. Most studies on the potential applications of SCGs have explored the potential of lignin as a fuel source to meet the rising global energy demands [69,71]. Lee et al. demonstrated that an organosolv pretreatment process enabled the utilization of SCG-derived lignin [72]. However, the bioprocessing of lignin is not cost-effective due to the high recalcitrance of this compound [68]. Cellulose and hemicellulose from SCGs can be hydrolyzed to form fermentable monosaccharides, particularly mannose and galactose, after which these compounds can be further utilized as a substrate for microbial fermentation and the production of other valuable chemicals such as polyhydroxyalkanoates [11,70]. Steam pretreatment is an effective method for the solubilization and reduction in enzyme dosage prior to the enzymatic hydrolysis of SCGs [73]. However, steam pretreatment at 175 °C strongly promotes the formation of sugar degradation products such as hydroxymethylfurfural and furfural, which inhibit microbial activity in subsequent fermentation processes [74]. The toxic degraded products present in SCG hydrolysates could have antimicrobial activity and, therefore, dewnstream detoxification steps may require increased coffee waste utilization [75]. Moreover, the microbial fermentation of fermentable sugars recovered from SCG using different strains can yield a wide range of products, such as bioethanol, biogas, poyhydroxyalkanoate polymers, and high-value platform chemicals [76]. Table 1 presents a collation of the level of different chemical components in coffee byproducts generated during coffee processing.

## 3. Value-Added Products from SCG

The processing of coffee generates varieties of by-products from different coffee processing steps as shown in Figure 2. As stated before, SCGs are the major by-product of coffee, and produce wide varieties of value-added products: biofuel, biosugar, bio-oil, bioactive compound, enzymes and organic acids, biopolymers, carotenoids, biosorbents, antioxidants, and biocomposites.

### 3.1. Biofuels

Due to their high carbohydrate contents, SCGs are considered a uniquely well-suited raw material for the production of biofuel, particularly bioethanol. Similar to other biomass materials used in bioethanol production, SCGs must be pretreated to enhance saccharification efficiency due to the complex components in their lignocellulosic structure. Although there are many types of pretreatment methods, such as those using diluted acid, ammonia, supercritical CO_2_ extraction, ionic liquid, and alkaline solutions [14,81], only a few of these methods are potentially feasible. Among them, dilute acid pretreatment results in high monosaccharide concentrations (58.4 g/L) and a high ethanol conversion yield (0.46 g ethanol per gram of total fermentable saccharides) after fermentation by *Saccharomyces cerevisiae* [93]. Moreover, many other studies have reported high ethanol production rates (> 50% conversion yield) from SCGs pretreated with dilute acid [84,94]. The obtained bioethanol can be converted into other biofuels, such as ethyl tert-butyl ether (ETBE), which is designed for use in EU gasoline markets, or high octane iso-octane [95,96,97,98]. Another inexpensive but effective method is alkaline pretreatment, in which a highly concentrated alkaline (sodium hydroxide) solution is used to produce polysaccharides. This results in carbohydrates that can be further processed, into ethanol [99]. This approach is highly feasible because it involves an easy purification process and renders high-purity products. A method known as popping pretreatment has been proven to enhance the saccharification of SCG (the ethanol concentration and yield were 15.3 g/L and 87.2%, respectively) [80], as well as other kinds of lignocellulosic biomass for bioethanol production [29,100,101,102].

Although SCGs are used for biodiesel production, the extracted SCG oils can also be converted into biodiesel using chemical and biocatalytic methods coupled with bases (NaOH and KOH) and acids (H_2_SO_4_ and HCl) as chemical catalysts. In previous studies, maximum biodiesel yields of 73% were achieved using NaOH [103]. In contrast, a final biodiesel yield of 96 wt% was obtained using KOH [104], and this yield could be further enhanced to nearly 100% if the process was conducted at a higher temperature (70 °C) [105]. In fact, the production of SCGs-derived biodiesel is highly feasible, representing 3.5% (0.9 million tones out of total biodiesel production of 26 million metric tons in 2014) of the worldwide biodiesel market share [106]. Recently, several studies have described the in situ preparation of biodiesel from SCGs, which contain 10–15% oil, depending on the coffee species [105]. Solvent extraction is commonly used to recover oil from SCGs. Biodiesel was synthesized from oil extracted from SCGs via transesterification between triglycerides and lower alcohols [105]. To eliminate the need for a separate oil extraction process, in situ transesterification, a process of simultaneous extraction and transesterification steps, was applied to produce biodiesel [107]. However, this process demands high energy consumption for recovering excess methanol; therefore, the process is not economically feasible to produce biodiesel from SCG [108]. Another challenge of transesterification of SCG oil arises due to its high free fatty acid (FFA) content, which reacts with alkali catalysts to form soap [107]. Therefore, SCG oil with lower FFA levels should be used to avoid excess saponification and the deactivation of the alkaline catalyst during the transesterification process [109]. Washing SCG with methanol was also reported to reduce its FFA content [107]. Moreover, Caetano et al. proposed a two-step procedure of acid esterification followed by alkaline transesterification for SCG oil with high FFA, resulting in higher biodiesel yields compared to the direct transesterification of SCG [82]. At the pilot-scale, this approach achieved SCG-to-biodiesel conversion yields of up to 83% [107].

In addition to basic products, such as bioethanol, recent studies explored the potential of SCGs as a raw material for the production of other value-added products. This involves the development of an integrated process in which both bioethanol and other high-value-added products, such as biosugars, are produced. These processes are designed to meet the requirement of greener processes, in which pretreatment and saccharification processes are included to achieve high quality products (Figure 3). It is noteworthy that the price of bioethanol has not always been lucrative for business. As a result, the production of biosugars, which are products of saccharification that are not used for fermentation, seems to be a suitable solution to utilize SCGs for large-scale industrial operations. SCGs can be used to produce bioethanol and biosugars, such as mannose and manno-oligosaccharides (Figure 3). A mass balance analysis demonstrated that approximately 15.7 g DW of D-mannose and 11.3 g DW of ethanol could be produced from 150 g DW of ethanol-pretreated SCGs [29]. In another study, 3.1 g of bioethanol and a significant amount of mannose-oligosaccharides and D-mannose were produced from 100 g DW SCGs after incorporating delignification and defatting pretreatment steps [30]. This integrated process provides a promising approach for extending the applications of biorefineries at a commercial scale using lignocellulosic biomass as a raw material.

### 3.2. Bio-Sugars

High-value biosugars are promising new products derived from lignocellulosic biomasses and wastes, which have recently received substantial investment from large companies [110,111,112,113]. Although previous studies have described physical and chemical methods for the production of biosugars, enzymatic conversion processes have become the preferred approach because they are stable, safe, environmentally friendly, and highly effective for biosugar production. Due to the massive global availability of SCGs and their high content of mannose/mannan, SCGs are an excellent resource for producing D-mannose and manno-oligosaccharides. However, these resources continue to be grossly underutilized. A composite process consisting of several simultaneous steps including pretreatment with ethanol, enzymatic hydrolysis via high efficiency in-house cellulase and pectinase, fermentation, color removal using activated carbon, and pervaporation was previously developed to produce D-mannose and bioethanol [29]. The entire method can be considered a green process due to the exclusive use of nontoxic chemicals and the constant recycling of ethanol. An upgraded process was later developed to produce D-mannose, manno-oligosaccharides, and bioethanol in a continuous process, which incorporated another pretreatment method including delignification and defatting steps to almost fully eliminate non-saccharides and lipids from the SCGs. The mannose-oligosaccharide, D-mannose, and bioethanol production yields from 100 g DW SCGs were 15.9, 25.6, and 3.1 g, respectively [30]. Unlike other products and applications of SCGs, these two processes and products can be extended to produce high value-added products such as biosugar.

### 3.3. Bio-Oils

SCGs can also be used for the production of bio-oil as coffee cherries typically have a fatty acid content of approximately 15%. However, this varies depending on the type of coffee. Simpler extraction techniques, such as conventional solvent (hexane or diethyl ether) extraction via the Soxhlet method or supercritical extraction, are required for bio-oil extraction [104,114]. In one study, a 6% yield of coffee oil was obtained with hexane under reflux conditions, whereas 14% was extracted via the Soxhlet extraction method [115]. However, in our previous study, the extraction of SCGs using hexane as a solvent resulted in a coffee oil yield of approximately 9.7% [116]. Moreover, up to 15.4% DW of oil was extracted from SCGs using a supercritical carbon dioxide (scCO_2_)-based approach [117]. Additionally, the hydrothermal liquefaction (or fast pyrolysis) of SCGs can produce coffee oil with a conversion yield of 66% at 630 °C [118]. The resulting coffee oil can be used as transportation fuel, among other applications [119].

### 3.4. Bioactive Compounds

Coffee contains a large number of bioactive substances with antioxidant, hypolipidemic, hypoglycemic, neuroprotective, and other biological properties [78]. Moreover, SCGs are an excellent source of phenolic compounds such as chlorogenic and caffeic acids. These compounds are often extracted via solid–liquid extraction. However, they could also be extracted using the SSF method, which not only enables the extraction of high-quality bioactive compounds, but also eliminates the need for toxic organic solvents that are typically used in chemical and thermal methods [18,120]. Despite concerns regarding their eco-toxicity, these bioactive compounds have been proven to promote human health due to their antioxidant, antibiotic, anti-inflammatory, and hepatoprotective properties [121]. Ultrasound-assisted solid–liquid extraction is one of the most effective methods to improve the efficiency of phenolic extraction. This approach can achieve total phenolic yields of up to 3.6% [122], which represents a nearly six-fold increase in extraction compared to the above-described SSF extraction method. Chlorogenic acid and its derivatives (e.g., caffeoylquinic, feruloylquinic, p-coumaroylquinic, and quinic acids, and esters of caffeic and ferulic acids) are present in coffee beans and SCGs [18,120]. Using the solvent extraction method with aqueous ethanol solutions, Zuorro and Lavecchia were able to extract approximately 90% of phenolic extracts from SCGs [120]. The recovery of phenolic compounds can be further improved via process optimization. For example, a solid–liquid extraction method capable of recovering high phenolic yields were developed using methanol as a solvent. However, the applications of methanol are limited in the food industry and life sciences due to its toxicity [84,94].

In summary, the production of bioactive compounds from SCGs could be considered industrially and economically feasible due to the organic nature of the substrates for phenolic extraction. However, their applicability and success are still highly contingent customer acceptance.

### 3.5. Enzymes and Organic Acids

Several industrial enzymes, such as amylase, cellulase, protease, xylanase, and pectinase, have been produced from SCGs, particularly by SSF and submerged fermentation (SmF) [123,124,125]. Among these two methods, SSF is more widely used for enzyme production because this method offers operational and economic advantages over SmF. Using SSF, the yield of cellulase from SCGs reached up to 71% using *Paenibacillus chitinolyticus* [126]. Under SSF, xylanase was produced from lignocellulosic coffee waste by *Penicillium* sp. [124], pectinase was produced from SCGs by *Aspergillus* sp. [125], and β-fructofuranosidase was produced from coffee silverskin by *Aspergillus* sp. [127]. Instead of disposing of SCGs into the open environment, the continued improvement of enzyme production techniques for SCGs through decades-long research and development could enable the large-scale commercial production of enzymes from SCGs.

Another promising application of SCGs is for the production of organic acids such as citric and gibberellic acids. An estimated 1.5 g of citric acid per 10 g DW SCGs can be produced via the SSF approach with *Aspergillus niger* [128]. Another study reported that *Gibberella fujikuroi* could produce up to 492.5 mg of gibberellic acid per-kg SCGs [129]. Moreover, an integrated process allowing for the simultaneous extraction of chlorogenic acids and production of bioethanol from SCGs was demonstrated at pilot- and large-scales [130]. This is a feasible approach due to the high content of chlorogenic acids recovered from SCGs.

### 3.6. Biopolymers, Carotenoids, Biosorbents, Antioxidants, and Biocomposites

Many efforts have been made to produce value-added products such as biopolymers, biosorbents, polyphenols, and biocomposites from SCGs. Polyhydroxyalkanoates (PHAs) have gained increasing attention due to their potential applicability as suitable bio-based polymers due to their biodegradability and the easy manipulation of their thermoplastic and elastomeric properties. PHAs can be synthesized by *Bacillus megaterium* and *Burkholderia cepacia* from polyphenols extracted from SCGs via acidic hydrolysis [70]. Additionally, poly-(3-hydroxybutyrate) can be produced from SCGs oils by *Cupriavidus necator* H16 [131]. Moreover, liquid polyols, which are used for the industrial production of polyurethane foams [132], can be prepared from acid liquefaction of SCGs (sulfuric acid) using PEG 400 or glycerol as solvents [133].

Carotenoids are bioactive compounds that can be produced from the microbial fermentation of SCGs. These compounds are widely used in the food, pharmaceutical, and cosmetic industries due to their high antioxidant and antimicrobial capacity. These compounds can be produced using various microbes such as yeasts, filamentous fungi, bacteria, and algae, which utilize organic wastes as a carbon source. For example, hydrolyzate derived from SCGs can be used as a substrate by the carotenogenic yeast *Sporobolomyces roseus* for the production of carotenoids [131].

SCGs can be used as inexpensive bio-sorbents for the removal of dyes, heavy metals and pollutants during liquid waste treatments [10]. Furthermore, magnetic fluid treatment is an interesting development, in which magnetized SCGs are used as an adsorbent for the removal of water-soluble dyes such as crystal violet, malachite green, amido black 10 B, Congo red, Bismarck brown Y, acridine orange, and safranin O [134]. Similarly, various heavy metals such as Cd(II), Cu(II), Pb(II), Cr(VI), Ni(II), and Zn(II) can also be removed using SCGs and other coffee wastes pretreated with sodium hydroxide [135]. Another study demonstrated that increasing the roasting temperatures of coffee above 170 °C significantly enhanced the adsorptive properties of SCGs [136]. Furthermore, activated carbon produced from the thermal treatment of SCGs was also reported to be an efficient adsorbent [137]. Additionally, biocomposites prepared from renewable, recyclable, and biodegradable materials are becoming increasingly popular due to their sustainability, in addition to promoting a circular economy. For example, Lee et al. described a method for the fabrication of polyvinyl alcohol/SCG nanocomposites and their tensile strengths were comparable to those of polyvinyl alcohol/carbon black nanocomposites [72]. Baek et al. demonstrated the preparation of green composites, in which natural fillers such as SCGs and bamboo flour were mixed with polylactic acid, after which a coupling agent was used to enhance the bonding between the natural fillers and polymers [138]. These composites could be used in a variety of products, such as interior decorating materials, as well as in the construction industry (Figure 2). Table 2 shows the utilization of different components of spent coffee waste as a green bio-waste source in the various industry.

### 3.7. Photothermal Materials, Catalyst for Creation of Nanoparticles, and Synthetic Leather

A novel application of SCGs as photothermal materials has been recently reported by Chien et al. [147,148]. Photothermal materials can convert light energy into thermal energy and are widely used in the areas of energy and biomedicine. The SCGs generated heat through NIR irradiation were effective for eliminating planktonic bacteria and biofilms.

A recently published paper employed SCGs to synthesize silver nanoparticles as catalyst [149]. In this research, silver nanoparticles attached on the surface of SCG powders immobilized in poly(ethylene terephthalate (PET) sheets were synthesized. The catalyst displayed good catalytic activity for the reduction of 4-nitrophenol in the presence of sodium borohydride (NaBH_4_), with excellent durability of maintaining >90% conversion for at least seven recycles. SCGs can be also used as catalyst for both biodiesel and hydrogen production. Atelge synthesized a heterogeneous catalyst derived from SCGs for transesterification reaction [150]. Their results revealed that the produced catalyst was used five times with 91.57% biodiesel yield under 9:1 alcohol and oil ratio, 3 wt% catalyst loading and 90 °C reaction temperature. Moreover, the methanolysis reactions of NaBH4 was achieved 100% hydrogen conversion ratio.

Tian et al. recently developed the preparation method of water-based synthetic leather by reusing SCGs as fillers [151]. According to this study, sustainable coffee-ground synthetic leather fully met the performance of aqueous synthetic leather for apparel and luggage.

## 4. Conclusions

This review discussed the broad applications of coffee by-products collected throughout the farming, processing, and consumption of coffee. Instead of being discarded as waste, coffee by-products can be used as raw materials for the production of a variety of value-added products. For example, profitable and health-promoting products can be manufactured using coffee leaves and flowers, as well as the pulp, husks, silverskin, and SCGs derived from coffee processing and brewing. SCGs, an abundant food waste worldwide, can be used as a starting material for the production of multiple products such as phenolic compounds, polysaccharides, biodiesel, antioxidants, bio-oil, and bioethanol. Biopolymers such as polyhydroxyalkanoates can also be produced from SCGs. However, pretreatment of SCGs is essential in some processes such as the production of bio-ethanol and biogas to ensure the optimal enzymatic hydrolysis of the cellulose present in the SCGs due to the high levels of anti-microbial compounds in coffee waste. Among the pretreatment methods examined thus far, acid hydrolysis is a promising approach not only for the recovery of sugars from SCGs but also lipids.

The conversion of coffee waste to bioethanol and biosugars represents a promising means to meet the growing demand for biofuels and the strict regulation of environmental pollution from factory emissions. Moreover, in addition to its environmental benefits, the utilization of coffee by-products could substantially increase the profits of the coffee processing industry. To take advantage of these by-products in the most efficient manner, large-scale biofuel and biosugar production units could be directly incorporated into coffee processing plants in the future. However, cost-effective, practical, and innovative inventions are still necessary for such units to enhance the sustainability of the coffee processing business.

## Figures and Tables

**Figure 1 molecules-28-03562-f001:**
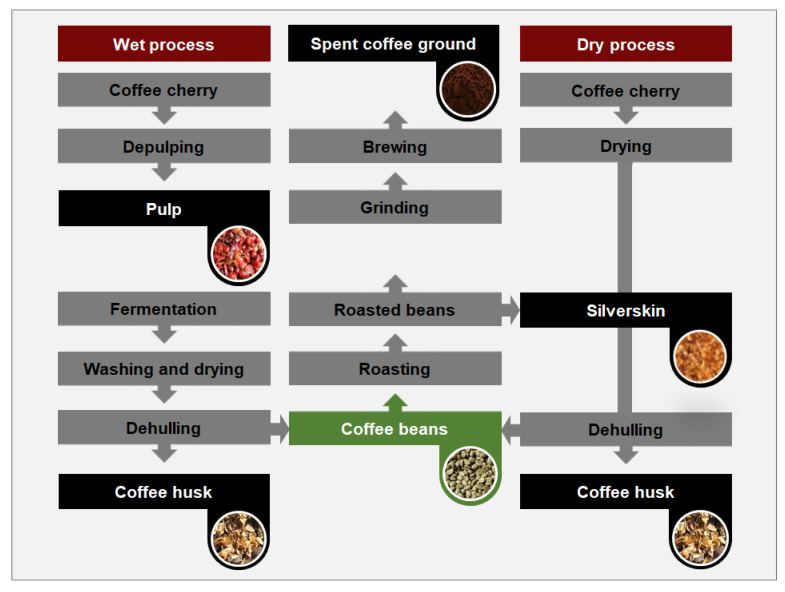
By-products generated during wet and dry processing of coffee.

**Figure 2 molecules-28-03562-f002:**
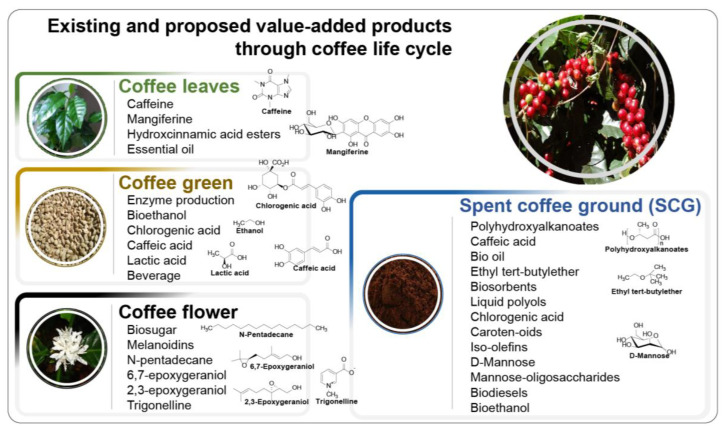
Existing and proposed value-added products throughout the coffee life cycle.

**Figure 3 molecules-28-03562-f003:**
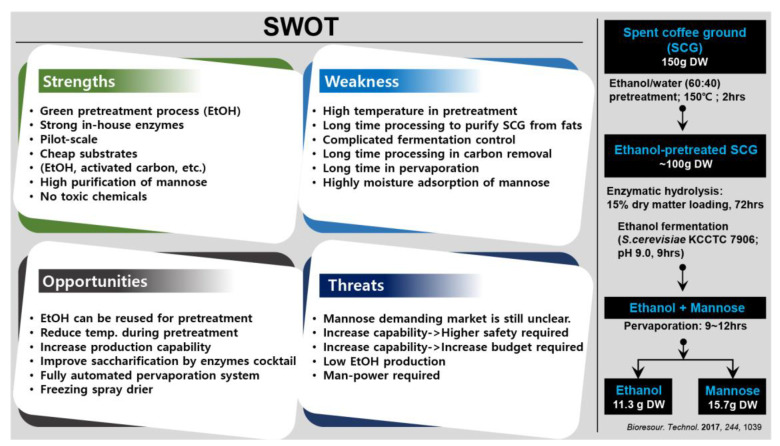
Schematic diagram for the production of biosugars and bioethanol from spent coffee grounds (SCGs) and the economic point-of-view [29].

**Table 1 molecules-28-03562-t001:** Collation of the levels of different chemical components in coffee byproducts generated during coffee processing.

Chemical Component	Green Coffee	Coffee Pulp	Coffee Husk	Silverskin	Roasted Coffee	Spent Coffee Ground	Coffee Flower	Coffee Leaf	Ref.
Carbohydrates	60.0	44.0–55.0	57.8	44.0	58.5	60.3–82.0		51.0–63.9	[11,77,78,79]
Cellulose	18.0–65.0	9.18–63.0	39.0–61.0	16.0–46.0	38.0–41.5	8.6 −47.3	3.1 −14.6	2.4	[11,29,30,39,40,62,78,80,81,82,83,84,85,86]
Hemicellulose	3.0 −15.0	2.0 −66.0	4.0 −10.0	4.0 −22.0	24.0–39.1	32.0–43.0	-	-	[11,61,78,81,84,85,86,87]
Xylose	-	-	-	4.7–7.6	-	0.3–1.1	2.4	2.7	[29,30,39,40,62,80,83,84,88]
Arabinose	20–35	-	-	2.0–3.5	0.1	1.7–3.6	0.3–3.8	3.5	
Mannose	10–20	-	-	1.8–2.6	-	19.1–21.6	0.2–1.3	0.6	
Galactose	55–65	-	-	3.8	-	8.2–16.4	2.7	1.0–2.3	
Rhamnose	-	-				0.1	0.7	-	[39,80]
Lignin	1.0–5.6	12.2–22	9.0	1.0–39.0	5.8–44.8	23.9–33.6	-	-	[29,30,61,81,82,85,86,89,90]
Insoluble lignin	-	-	-	21.0		17.6–31.9	-	-	[29,30,69,82]
Soluble lignin	-	-	-	7.6		1.7–6.3	-	-	
Lipids	8.0–18.0	0.3–2.5	0.5–6.0	0.3–4.0	11.0–17.0	6.0–38.6	-	-	[11,77,84,85,89]
Proteins	8.5–13.4	4.4–12.0	3.0–13.0	15.0–23.0	3.1–17.4	11.5–18.0	6.5–9.1	14.4–19.0	[11,39,40,63,77,78,79,81,82,84,85,89,91,92]
Ash	3.0–5.0	5.4–15.4	6.0	4.7–8.0	1.3–4.3	1.1–2.2	7.5–8.1	8.8–12.4	
Caffeine	0.8–4.0	0.8–5.7	0.5–2.0	0.0–1.4	1.0–2.4	0.02–0.4	0.9–1.1	1.6–2.5	[11,32,39,40,77,78,79,83,84,85,86,91]
Tannins	-	1.8–8.6	4.5–9.3	0.02	-	0.02	-	-	[11,86]
Chlorogenic Acids	3.8–10.0	1.0–10.7	2.0–12.6	3.0–15.8	0.9–8.3	1.8–11.5	1.3	-	[11,40,77,84,86]
Pectins	2.0	4.4–12.4	0.5–3.0	0.02	2.0	0.01	-	-	[78,85,86]

**Table 2 molecules-28-03562-t002:** Utilization of different components of spent coffee waste as green bio-waste sources in various industries.

Active Compounds from Spent Coffee Ground	Products	Role/Function	Application	Ref.
Bio-oil	Coffee oil makeup remover	Cleansing agent	Cosmetic	[139]
Bio-oil	Bio-polymer: poly(3-hydroxybutyrate) (PHB)	Biodegradable plastic	Packaging material	[131]
Hemicellulose/Cellulose	Bio-sugars (mannose, galactose, arabinose, and glucose)	Mannitol and fermentation feedstock	Chemical and food industry	[140]
Lipids	Ethanol, biodiesel	Biofuel/Alternative energy	Biorefinery/Transportation	[94]
Lignin	Hydrolyzed spent coffee grounds	Antioxidant: protect lipid oxidation	Biomedical/Industrial	[141]
Lignin	Bio-sugars (D-mannose, manno-oligosaccharides)	Biofuel feedstock	Value added biorefinery product	[117]
Lignocellulose	Bioethanol	Biofuel	Value-added biorefinery product	[50]
Lignocellulose	Xylanase	Xylan biodegradation	Biopulping, prebleaching of Kraft pulps, clarifying fruit juices and wine	[124]
Lignocellulose	glucose, galactose and mannose	Production of biofuels, amino acids and enzymes	Nutraceutical and food product	[142]
SCG (Ohmic heating extraction)	Dietary fiber bound with antioxidant	Antioxidant/anti-diabetic	Antidiabetic bakery product	[143]
Untreated SCG	Magnetically modified SCG	Biobsorbants	Xenobiotic dye removal	[134]
Untreated SCG		Substrate for cultivation of edible fungus	Mushrooms production	[2]
Untreated/Whole SCG	Alcoholic beverages rich in ester and higher alcohol	Fermentation substrate for fermented and distilled beverages	Product diversification/Novel product development	[144]
Wet SCG	Biodiesel (Methanol)	Alternative energy	Biofuel production	[145]
SCG (Ultrasonic extraction)	Phytosterols (β-sitosterol campesterol, stigmasterol, cycloartenol)	Bioactive compound	Nutraceutical and cosmetic	[146]
Delignified and defatted SCG	D-mannose, manno-oligosaccharides, and bioethanol	Bio-sugar	Value-added biorefinery product	[30]
SCG (Ultrasonic)	Phenolic compounds (chlorogenic and protocatechuic acids)	Bioactive compounds	Biomedical and food	[123]
SCG (Pre-activated)	Cellulase (immobilized)	Cellulose depolymerization	Biofuel and food and pharmaceutical	[126]

## Data Availability

Not applicable.

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
