# Peer review of "Value-Added Products from Coffee Waste: A Review"

_molecules, 2023, doi:10.3390/molecules28083562_

Round 1
Reviewer 1 Report
The manuscript is an important addition to the exisiting data about the coffee waste valirization.
The manuscript can be accepted for publication after revision.
Please add the data from Trchouian group about coffee wastes especially SCG and coffeesilverskin and also add possible bioenergy production from coffee waste.
Reviewer 2 Report
The authors described in their narrative review the value-added products from coffee production waste. The manuscript is well written and covers all scientifically important questions considering use of coffee production waste.
I have only several minor suggestions:
The sentence lines 9-11 is too long and it could be shorten especially as it is the first sentence of the manuscript, it should be more effective if the authors rewrite it.
Figure 1. title should be corrected.
Line 217. There is a typing mistake organosolv should be changed to treatment with organic solvents or similar.
Figure 2 and 3. The quality of pictures that can be downloaded as supplementary material is good, but their visibility in the manuscript template should be improved.
Line 376. Please check whether there is a description for abbreviation SSF when it is used for the first time in the text.
The authors could add some pictures of chemical structures of substances present in high amounts in waste of coffee production
The title of the Manuscript could also be modified as value-added products not only from spent coffee waste are described in the review
Reviewer 3 Report
Review of molecules-2298100
This is a nice review of the utilization of coffee waste (leaves, flowers, spent coffee grounds, and so on). The writing is systematic and easy to follow. There are however several issues to be corrected in order to improve the manuscript even more, such as:
1. Section 3: The discussion of the value-added products from spent coffee ground (SCG) is not comprehensive yet. Please include these references, since coffee grounds can be utilized for the synthesis of advanced materials for environmental sustainability (and not just burned for energy or bulk chemicals/feedstock chemicals), such as:
· As photothermal materials --> Journal of the Taiwan Institute of Chemical Engineers 137 (2022) 104259 https://doi.org/10.1016/j.jtice.2022.104259
· As photothermal materials --> Journal of Chemical Environmental Engineering 10 (2022) 107131 https://doi.org/10.1016/j.jece.2022.107131
· As catalyst for creation of nanoparticles--> Food and Bioproducts Processing 121 (2020) 193-201 https://doi.org/10.1016/j.fbp.2020.02.008
· As synthetic leather --> Sustainability 14:21 (2022) 13971 https://doi.org/10.3390/su142113971
2. Line 241: …wide varieties..
3. Line 293, Figure 3: Please write scientific names in italic. --> S. cerevisiae
4. Line 294, caption of Figure 3: Please add the information of the reference (Bioresource Technology 244 (2017) 1039 in the figure caption.
5. Line 248: ...efficiency due to the… --> delete the dot, merge the two separated sentences.
6. Reference 31: Please write scientific names in italic --> Coffea arabica
7. Reference 33: Please write scientific names in italic --> Coffea
8. Reference 34: Please write scientific names in italic --> Coffea arabica
9. Reference 36: Please write scientific names in italic --> Coffea arabica
10. Reference 37: Please write scientific names in italic --> Coffea arabica
11. Reference 38: Please write scientific names in italic --> Coffea arabica
12. Reference 43: 3rd --> superscripted th
13. Reference 44: Please write scientific names in italic --> Pleurotus ostreatus
14. Reference 44: Please write scientific names in italic --> Pleurotus pulmonarius
15. Reference 45: Please write scientific names in italic --> Acetinobacter sp.
16. Reference 46: Please write scientific names in italic --> Bacillus subtilis
17. Reference 53: Please write scientific names in italic --> Coffea arabica
18. Reference 82: Please write scientific names in italic --> Coffea robusta
19. Reference 86: Write the journal name in the font type and font size required by MDPI
20. Reference 89: Write the journal name in the font type and font size required by MDPI
21. Reference 97: Please write scientific names in italic --> Citrus unshiu
22. Reference 121: Please write scientific names in italic, with uppercase letter for the genus --> Penicillium sp.
23. Reference 123: Please write scientific names in italic --> Paenibacillus chitinolyticus
24. Reference 124: Please write scientific names in italic --> Aspergillus japonicas
25. Reference 125: Please write scientific names in italic --> Aspergillus niger
26. Reference 140: Please write scientific names in italic --> Coffea arabica
Reviewer 4 Report
The study proposed by the authors presents an interesting area of both food technology and waste management, which causes interest for both scientists and practitioners.
The undoubted advantage of the manuscript is the specific goal, the use of study subject that is considered as by-product/ waste in coffee production and interesting and reliable introduction of the presented research.
I am satisfied with the current form of presented publication, which is interesting and written in a good style.
Round 2
Reviewer 3 Report
Review of molecules-2298100-v2
The manuscript has been improved really well, it can be accepted in its current form.